# Kinetics of Excretion of the Perfluoroalkyl Surfactant cC_6_O_4_ in Humans

**DOI:** 10.3390/toxics11030284

**Published:** 2023-03-20

**Authors:** Silvia Fustinoni, Rosa Mercadante, Giorgia Lainati, Simone Cafagna, Dario Consonni

**Affiliations:** 1Toxicology Lab, Department of Clinical Sciences and Community Health, University of Milan, 20122 Milano, Italy; 2Fondazione IRCCS Ca’ Granda Ospedale Maggiore Policlinico, 20122 Milan, Italy

**Keywords:** serum cC_6_O_4_, half-life, elimination route, biomonitoring, urine

## Abstract

cC_6_O_4_ is a new-generation perfluoroalkyl surfactant used in the chemical industry for the synthesis of perfluoroalkyl polymers. It was introduced as a less biopersistent substitute of traditional perfluoroalkyl surfactants such as PFOA, but its kinetics in humans was never investigated. This work is aimed to investigate the kinetics of elimination of cC_6_O_4_ in exposed workers. Eighteen male individuals occupationally exposed to cC_6_O_4_ in the production of fluoropolymers volunteered for the study. Blood and urine samples were collected from the end of a work-shift for the following 5 days off work. Serum and urinary cC_6_O_4_ were measured by LC-MS/MS. Seventy-two samples with serum cC_6_O_4_ ranging from 0.38 to 11.29 µg/L were obtained; mean levels were 3.07, 2.82, 2.67 and 2.01 µg/L at times 0, 18, 42 and 114 h. Two hundred and fifty-four urine samples with cC_6_O_4_ ranging from 0.19 to 5.92 µg/L were obtained. A random-intercept multiple regression model was applied to serum data and a half-life of 184 (95% CI 162–213) h for a first-order kinetics elimination was calculated; a mean distribution volume of 80 mL/kg was also estimated. Pearson’s correlation between ln-transformed serum and daily urine concentrations was good, with r ranging from 0.802 to 0.838. The amount of cC_6_O_4_ excreted daily in urine was about 20% of the amount present in serum. The study allowed calculating a half-life for cC_6_O_4_ in blood of about 8 days in humans, supporting its much shorter biopersistence in comparison with legacy PFAS. The good correlation between urine and serum cC_6_O_4_ suggests urine as a possible non-invasive matrix for biomonitoring. The amount of cC_6_O_4_ excreted daily in urine suggests urine as the sole elimination route.

## 1. Introduction

cC_6_O_4_ (acetic acid, 2,2-difluoro-2-[[2,2,4,5-tetrafluoro-5-(trifluoromethoxy)-1,3-dioxolan-4-yl]oxy]-, ammonium salt (1:1) CAS 1190931-27-1) (see Figure 1) is a perfluoroalkyl surfactant used in aqueous emulsions for the synthesis of plastomer and elastomer fluoropolymers in the chemical plant of Solvay Specialty Polymers Italy S.p.A., located in Spinetta Marengo (AL), Italy.

This chemical was developed as a new polymerization surfactant to overcome the hazard issues associated with perfluorooctanoate (PFOA), mostly biopersistency. Indeed, toxicokinetics studies in rats provided evidence of a short half-life in serum (4 to 7 h), with the majority of the excretion via urine occurring usually within 24 h, and no evidence of biotransformation and bioaccumulation [1]. According to CLP (Regulation (EC) No. 1272/2008), cC_6_O_4_ was classified as harmful if swallowed (H302), able to cause skin irritation and serious eye damage (H315 and H318); moreover, it may cause damage to organs (liver) through prolonged or repeated exposure (oral) (H373). The substance does not accumulate nor exert toxic effects in the aquatic organisms; it is not a persistent bioaccumulative toxic (PBT) or a very persistent very bioaccumulative (vPvB) chemical on the basis of the Annex XIII REACH. cC_6_O_4_ has been produced since 2012 and it was introduced to replace PFOA; today it is the candidate for the substitution of other polymerization surfactants in chemical processes and used as a bridging solution to a Non FluoroSurfactant technology.

Human biological monitoring of exposure is a powerful tool to assess chemical exposure, integrating all exposure routes and sources, and taking into consideration characteristics of individuals [2]. Given the metabolic stability of perfluoroalkyl substances, biomonitoring is performed by measuring the unchanged chemicals in serum. Extensive literature is available on PFAS, with biomonitoring data in occupationally exposed individuals and in the general population, particularly regarding legacy PFAS, such as PFOA and perfluorooctane sulfonate (PFOS) [3]. Conversely, limited biological monitoring information has been reported on cC_6_O_4_. Particularly, data of workers of an Italian plant located in the Veneto region were retrieved in a technical report, with cC_6_O_4_ serum concentrations ranging from 0.5 to 932 µg/L [4]. More recently, a summary of 9 years of biomonitoring surveys in the chemical plant of Spinetta Marengo and in the research center of Bollate, both belonging to Solvay Specialty Polymers Italy SpA, Italy, was reported, with a large percentage of samples not quantifiable (below the analytical lower limit of quantification) and concentrations in the range of 0.1–56 μg/L [5].

The present work reports the results of a study on the kinetics of elimination of cC_6_O_4_ in occupationally exposed individuals. With this aim, blood and urine samples were collected immediately after a work-shift and during a period of 114 h while off work. The half-life of cC_6_O_4_ in serum and correlation between urine and serum concentrations were calculated, and a mass balance was performed by comparing the amount of cC_6_O_4_ excreted in urine and circulating in serum.

## 2. Materials and Methods

### 2.1. Field Study

The surveyed workers were employees of the chemical plant of Spinetta Marengo (AL), where cC_6_O_4_ is synthetized since 2012 and used as surfactant for the synthesis of fluorinated plastomers and elastomer polymers.

In this plant, the biological monitoring of cC_6_O_4_ started in 2013 and was performed annually (typically in February) in the frame of the occupational health and safety surveillance, according to the Italian law for the safety and health at workplaces [6], under the responsibility of occupational plant physicians; the workers signed an informed consent to agree to the survey [5].

The kinetics study was performed in collaboration with the occupational health physicians and industrial hygienists of the plant. From the historical data of the biological monitoring, it was observed that the highest exposure to cC_6_O_4_ was found in workers of the plastomers area. For this reason, the study was proposed to these workers. Eligible criteria were the following: being male; not affected by chronic diseases, particularly metabolic, hepatic and kidney diseases; and having a body mass index lower (BMI) than 30 kg/m^2^. Among eligible male workers, all volunteered to participate in the study.

The sample collection lasted 114 h, during which 4 blood samples and a variable number of urine samples were collected.

For blood, the sampling protocol was as follows:Day 1: A working day with a 6:00 to 14:00 work-shift; the first blood sample was collected at the end of the shift (time 0 h);Day 2: A day off; at 8:00 the second blood sample was collected (time 18 h);Day 3: A day off; at 8:00 the third blood sample was collected (time 42 h);Day 4 and day 5: Two days off; no blood sample was collected;Day 6: A working day with a 6:00 to 14:00 work-shift; before the start of the shift the fourth blood sample was collected (time 114 h).

About 5 mL of blood was drawn from the cubital vein in the infirmary of the site, by health personnel, using a tube with heparin as anticoagulant. Samples were centrifuged on site to separate serum, within 1 h after collection. Serum samples were frozen at −20 °C.

For urine, a sampling kit, including 25 numbered sterile recipients for urine (100 mL) and a void collection diary, was given to each participant with instructions for urine collection. The subject was asked to collect all urine of all voids starting from day 1 at 14:00 until day 3 at 8:00 (collection time 42 h). An additional sample was collected on day 6, before re-starting the shift, from most subjects (15 out of 18). Urine was self-collected and recorded in the form, indicating the void date, time and the recipient number/s. Initially, samples were kept at home, stored outside at ambient temperature, until the next morning, when they were given to the health personnel in the infirmary of the plant. Once there, urine samples were frozen at −20 °C.

A self-administered questionnaire was used to collect personal information such as height, weight, smoking habit and drug consumption.

The study was approved by the ethical committee of the University of Milan; each volunteer received the study protocol and signed the informed consent form.

The sample collection was performed from November 2021 to April 2022.

### 2.2. Analytical Measurements of Serum and Urinary cC_6_O_4_

The analytical work was performed following a previously published method with few modifications [7]. Briefly, each properly stored sample was thawed at room temperature. The sample was vortexed and an aliquot (20 µL of plasma or 40 µL of urine) was added with methanol (70 µL for plasma and 150 µL for urine) and 10 µL of the internal standard methanol working solution to obtain a 1:5 diluted sample containing 0.15 µg/L of internal standard (perfluoro-n-[1,2,3,4,6-13C5]hexanoic acid, chemical purity >98%, isotopic purity 99% Wellington Laboratories, Guelph, Canada). Plasma proteins were precipitated, and the clear supernatant was transferred to an autosampler vial; the same process was applied to urine samples. The injection volume was 10 µL. The chromatographic run was performed with an Agilent 1260 liquid chromatograph equipped with an Acquity HSS T3 C18 analytical column (2.1 × 100 mm, 1.8 µm, Waters) preceded by a SecurityGuard C18 pre-column (4 × 3 mm Phenomenex). A Hypersil GOLD delay column (3 × 50 mm, 3 µm, Thermo Fisher Scientific) was also installed between the pump and the injector. The separation was obtained with a binary pump linear gradient: mobile phases were aqueous 10 mM ammonium acetate buffer, pH 4.5 (eluent A), and acetonitrile (eluent B) pumped at a constant 0.2 mL/min flow rate. Detection was performed using a 5500 QTRAP Sciex triple quadrupole mass spectrometer equipped with an electrospray ionization source (ESI), operating in the negative ionization mode. Analytical standards were re-tuned and flow re-injected, allowing mass spectrometry parameter optimization; target compounds were detected by extracted ion chromatograms of monitored MRM transitions. The quantification was performed with calibration curves deriving from calibration standards that covered the cC_6_O_4_ linearity range concentrations (lower and upper limits of quantifications (LOQ): LLOQ 0.02 µg/L, ULOQ 1.25 µg/L). The calibration curves were built using serum or urine when appropriate to properly control for the specific matrix effect. The method was validated according to international guidelines for bioanalytical methods [8,9,10]. The quality requirements assessed in our previous work on serum were also achieved for urine matrix. Effect of dilution of serum and urine samples with methanol was assessed and deemed insignificant, enabling sample dilution when concentration was above the linearity range.

Urinary creatinine (crea) was determined using Jaffe’s colorimetric method [11]. No criteria of acceptability based on urine dilution was applied.

### 2.3. Elimination Kinetics of Serum cC_6_O_4_ and Clearance

The one-compartment first-order kinetics model was applied to evaluate the elimination kinetics of serum cC_6_O_4_. The kinetic constant k (h^−^^1^) was derived as the slope of the linear regression obtained plotting the natural logarithm of the concentration of serum cC_6_O_4_ at any time C (µg/L) on the y axis vs. the time elapsed since t_0_, t (h) on the x axis, according to Equation (1).
ln(C) = −k t + ln(C_0_)(1)

The serum half-life of cC_6_O_4_ (t_1/2_) was obtained applying Equation (2), obtained when the concentration was half its initial value, and therefore C/C_0_ = 2.
t_1/2_ = ln 2/k(2)

Based on the 95% confidence interval of k, the 95% confidence interval of the t_1/2_ (h) was also calculated.

The daily renal clearance (Cl) was calculated applying the following equation:(3)$$\mathrm{Cl}\;(\mathrm{mL}/\min)=\frac{amount\;cC_{6}O_{4}\;in\;urine-day\;2\;(µg)}{cC_{6}O_{4}\;serum\;concentration-day\;2}=\frac{Cu\;(µg/L)\times U\;(L/day)}{Cs\;(\mu g/L)\times 1440\;min/day}$$
where 1440 is the daily minutes. Serum and urine concentration of cC_6_O_4_ (C_s_ and C_u_, respectively), as well as the urine flow rate (U), were based on day 2, as in this day all parameters were available.

The apparent distribution volume (Vd) was calculated as
(4)$$\mathrm{Vd}\;(\mathrm{mL})=\frac{t1/2\;(min)\times Cl\;(mL/min)}{ln2}$$

The daily renal clearance and the apparent distribution volume were also normalized for weight unit, according to the following equations:(5)$${\mathrm{Cl}}_{w}\;(\mathrm{mL}/\min\;\times \;\mathrm{kg})=\frac{Cl\;(mL/min)}{Body\;weight\;(kg)}$$
and
(6)$${\mathrm{Vd}}_{w}\;(\mathrm{mL}/\mathrm{kg})=\frac{Vd\;(mL)}{Body\;weight\;(kg)}$$

### 2.4. Statistical Analysis

The concentrations of cC_6_O_4_ in serum and urine at different sampling times were described using the mean, standard deviation, minimum, 25th percentile, median, 75th percentile and maximum values. No sample below the LLOQ was found.

The urinary concentration was calculated for each collection day separately as the sum of the amounts of cC_6_O_4_ in the different voids divided by the total urine volume. Given that only for day 2 the urine collection was lasting for 24 h, this daily concentration was considered the most reliable and it was used for estimating the daily renal clearance (Cl), the distribution volume (Vd) and the daily excretion rate.

In order to ascertain whether the workers collected all urine voids, we estimated the expected daily excretion rate of urinary creatinine Crea_exp_, according to the formula [12]:(7)$${\mathrm{Crea}}_{\exp}\;(g/\mathrm{day})=body\;weight\left(kg\right)\times 24\;(g/(day\times kg))$$

From the % ratio between measured and expected daily amount of urinary creatinine, the percentage of incomplete urine collection was estimated (results reported in Table 1). 

This percentage is, however, affected by a large degree of variability, and only when it falls outside the range of 60–140% should urine collection be considered unreliable [13].

To calculate cC_6_O_4_ serum half-life we used random-intercept linear regression models applied to all pooled data to take into account intra-subject correlation. The potential effect modification of half-life by age and BMI (as continuous variables) were analyzed by introducing product (interaction) terms in the regression models and by calculating the corresponding Wald tests.

Statistical analyses were performed with Excel (Microsoft), SPSS (IBM) and Stata 17 (StataCorp. 2021).

## 3. Results

### 3.1. Study Subjects

In Table 1, the summary of personal and work characteristics of study workers are reported. Briefly, 18 male workers volunteered in the study. Their mean age was 44 years, and their BMI was 26.1 kg/m^2^. Three subjects were current tobacco smokers, and one was an e-cig smoker. About 80% of them were assuming drugs (mostly for hypertension and hypercholesterolemia).

Workers were involved in the synthesis of plastomers, particularly of a melt-processable perfluoroalkoxy fluorocarbon resin (Hyflon^®^) and of poly-tetrafluoroethylene (PTFE) (Algoflon^®^).

Seventy-two samples of blood (four for subjects) and 254 urine voids were collected. The number of urine voids collected by each individual from 14:00 of day 1 to 8:00 of day 3 was 14 ± 4. Overall, 14 samples were missed during collection and five containers were delivered broken to the laboratory.

Through the calculation of expected daily excretion rate of creatine, the completeness of sample collection was estimated, returning a ratio of 77 ± 21%. This supports the missed samples (n = 14) declared by the volunteers and the delivered broken/empty containers (n = 5). Nevertheless, subjects with a percentage of urine collection below the normal range (60–140%) [13] were only four, with values ranging from 38 to 57%.

### 3.2. Biological Monitoring

In Table 2 the results of the concentration of cC_6_O_4_ in serum and urine samples, the last without and with creatinine correction, are reported for each sampling day.

#### 3.2.1. cC_6_O_4_ Urinary Excretion

Nadler’s equation was used to estimate the volume of blood in each study subject:Blood volume = 0.3669 × h^3^ + 0.03219 × bw + 0.6041(8)
where h and bw are subject’s height (cm) and body weight (kg). Serum volume was assumed to be equal to the volume of plasma and estimated as 55% of the volume of blood. The volume of urine of day 2 was measured as the sum of the different voids in the 24 h. From the concentration of cC6O4 in serum and urine, and their volumes, the amount of cC6O4 in each matrix was estimated. The ratio between the amount of cC6O4 in urine and the amount in serum (day 1 and day 2) was evaluated as the percentage of cC_6_O_4_ excreted in urine. The mean value and SD were about 20 ± 10% for both days 1 and 2 (see Table 3).

#### 3.2.2. Pearson’s Correlation and Linear Regression

The correlation analysis between the serum concentration of cC_6_O_4_ of day 1 or day 2 and the urine concentration of day 2, each transformed in the natural logarithm, was quite good, with Pearson’s r correlation coefficients ranging from 0.803 to 0.838. The correlation coefficients were similar using urinary concentration both corrected and non-corrected for creatinine. The linear regressions corresponding to these correlations are illustrated in Figure 2. The correlation was very good also using the concentration of urinary cC_6_O_4_ in spot samples collected at the time of each blood drawing, with Pearson’s r correlation coefficients ranging from 0.850 to 0.535 for corrected and non-corrected urinary concentration.

### 3.3. Kinetics of Elimination of cC_6_O_4_, Clearance and Distribution Volume

Considering day 1 (time 0 h) as the initial time t0, and the concentration of cC_6_O_4_ in serum of the day 1 sample as the initial concentration C0, the elimination kinetics of cC_6_O_4_ was investigated by applying a first-order model on natural logarithm (ln) transformed pooled data of all subjects. A mean half-life (t_1/2_) of 184 (95% CI 162–213) h or 7.7 (95% CI 6.7–8.9) days was calculated. No effect modification by categories of age (p-interaction = 0.42) and BMI (p-interaction = 0.17) on the kinetics parameters was found. The elimination of cC_6_O_4_ from serum is reported in Figure 3. Data from each individual are reported separately with red dots connected by red lines, while the linear regression and 95% confidence interval are reported in bold black continuous and dotted lines.

The renal clearance (Cl) and the apparent distribution volume (Vd) are reported in Table 4, also normalized for the individual’s body weight.

## 4. Discussion

In the present work, the half-life of cC_6_O_4_ in occupationally exposed male individuals was estimated. The amount of cC_6_O_4_ excreted unchanged in urine was also investigated to evaluate the relevance of urine as the excretion route.

For the recruitment of volunteers, we evaluated the results of the annual biomonitoring campaigns performed from 2013 in the plant of Spinetta Marengo [5]. We noticed that a significant proportion of serum samples was below the lower limit of quantification (1 µg/L). Considering that, in the present study, we aimed to also investigate urinary cC_6_O_4_, expected to be in lower concentration compared to serum, we recruited our volunteers among plastomer workers as those with the highest exposures. Moreover, we adopted an analytical assay with much higher sensitivity, with an LLOQ of 0.02 µg/L for cC_6_O_4_ in both matrices [5]. The quantification of cC_6_O_4_ in 100% of samples testifies the goodness of the adopted strategy.

Comparing results of serum cC_6_O_4_ reported here with the previous work [5], we note a lowering of concentrations. In particular, the median and the maximum were 4 and 65 µg/L in Plastomer 2, and 4 and 20 µg/L in Plastomer 1 departments in 2021, while in the present study the median and maximum are to 2.34 and 11.3 µg/L. This difference might be explained by the increased sensitivity of the analytical assay that allowed to include all subjects in the statistical analysis, i.e., also those with cC_6_O_4_ in serum <1 µg/L (three individuals), which would have been excluded in 2021.

The concentrations of cC_6_O_4_ in serum and urine (corrected and non-corrected for creatinine) were highly correlated with Pearson’s correlation coefficients r > 0.80 (Figure 2). This very good correlation suggests that biomonitoring of exposure to cC_6_O_4_ could be performed using urine instead of blood; this would be more convenient, as urine is non-invasive and can be collected autonomously, without the needs of health personnel. To further explore this possibility, we evaluated the correlation between cC_6_O_4_ in serum and in spot urine samples collected at the time of each blood drawing, obtaining a very good correlation when the concentration of cC_6_O_4_ was corrected for creatinine (Pearson’s r = 0.850). This further supports the use of urine samples for the biological monitoring of the exposure to cC_6_O_4_.

The kinetics work was performed on volunteers who had blood sampled at fixed times and voids of urine collected over a defined period. This study design is convenient because it allows monitoring exposure in real-world conditions. At the same time, some drawbacks are present, and these include:Different doses experienced by study individuals, resulting in a large variability in the concentration of the chemical in the investigated specimens;Difficulties in adhering to the study protocol, particularly in collecting all voids, resulting in missing voids or incomplete urine collection. Moreover, this is often associated with an under-reporting of missing samples.

In the study of the kinetics of excretion of cC_6_O_4_ using the linear regression between ln-transformed concentrations of cC_6_O_4_ vs. time, different exposures result in different intercepts, but the slope of the linear regression is not affected. As the half-life is calculated from the kinetic constant k (see Equations (1) and (2)), which is derived as the slope of the linear regression, we can conclude that kinetics parameters are not influenced by different exposures. In fact, individual concentration lines were remarkably similar (parallel) across subjects (Figure 3). The second issue, associated with missing urine samples, was faced by calculating the excreted creatinine, as summarized in Table 1 [10]; this allowed us to estimate a rate of urine collection of 77% and to identify four subjects exceeding the low boundary of tolerance range [60–140%] [13]. Overall, we evaluated as “good” the compliance of our volunteers in collecting voids, but we acknowledge that the amount of cC_6_O_4_ excreted in urine is underestimated by about 23%. Considering this, the percentage of cC_6_O_4_ in serum daily excreted in urine would be closer to 25% instead of the 20% observed (Table 3); similarly, the renal clearance would be increased from 0.41 mL/min to 0.50 mL/min (Table 4).

cC_6_O_4_ was developed and introduced in the polymerization process by Solvay Specialty Polymers Italy S.p.A. to find alternatives with a better (eco)toxicological profile than persistent perfluoroalkyl chemicals such as PFOA. A kinetics study performed in rats showed a short half-life of 4–7 h [1]. The present study found a half-life in male humans of 184 (95% CI 162–213) h. This value is higher than that found in rats, as expected, as the metabolism in rodents is much faster than in humans.

This is the first study assessing the kinetics of elimination of cC_6_O_4_ in occupationally exposed individuals. Some studies on newly synthetized perfluoroalkyl chemicals, such as perfluorobutyrate (PFBA) [14] and GenX, were performed [15,16]. Table 5 provides an overview of the results of these studies together with a selection of studies investigating the kinetics of PFOA. The half-life of PFBA was evaluated in nine workers removed from the workplace for more than 7 days and was estimated to be 72 h (95% CI 43–101) [14]. The kinetics of GenX was investigated in 18 workers that took 3-4 days off: the half-life was 81 ± 55 h (mean ± SD) [15,16]. Noticeably, the half-lives of these compounds were all found to be within a few days (3 to 7.7 days), while a half-life of 3.16–3.5 years was reported for PFOA [4,5,17].

The short half-life of cC_6_O_4_ and its urinary elimination are supported by a recent in vitro and in silico study evaluating the interactions of cC_6_O_4_ with renal basal and apical membrane transporters that may contribute to the reuptake from the tubule lumen. Results showed no significant interactions, suggesting that reabsorption from the proximal tubule is not likely to interfere with the urinary elimination of cC_6_O_4_ in humans [18].

Other kinetics parameters obtained in the present study, such as the distribution volume (Vd), calculated to be 6.6 L (mean value), can help in understanding the degree of distribution of cC_6_O_4_ in the body. Given that study subjects have a mean body weight of 79 kg, the volume of serum was estimated to be 3.1 L, the volume of the interstitial fluids to be 12.6 L (16% of body weight) and the total volume of body liquids to be 48 L (60% of body weight). Comparing these volumes with the Vd of cC_6_O_4_, it is possible to appreciate that this is more than twice the serum volume, suggesting that cC_6_O_4_ is not exclusively present in serum, but it is likely distributed also into other compartments. At the same time, given that the Vd is about 1/8 of the total volume of body liquids, we speculate that no accumulation in poorly perfused compartments and organs is present.

In conclusion, the present work evaluated the kinetics of excretion of cC_6_O_4_ in male workers. The half-life of 7.7 days, the daily urinary excretion of about 20% of the plasma dose, and the distribution volume supported the short biopersistence and the lack of bioaccumulation of cC_6_O_4_ in the human body. As this study does not includes females and younger or elder subjects, differences in the kinetics parameters could be present in individuals with different personal characteristics.

## Figures and Tables

**Figure 1 toxics-11-00284-f001:**
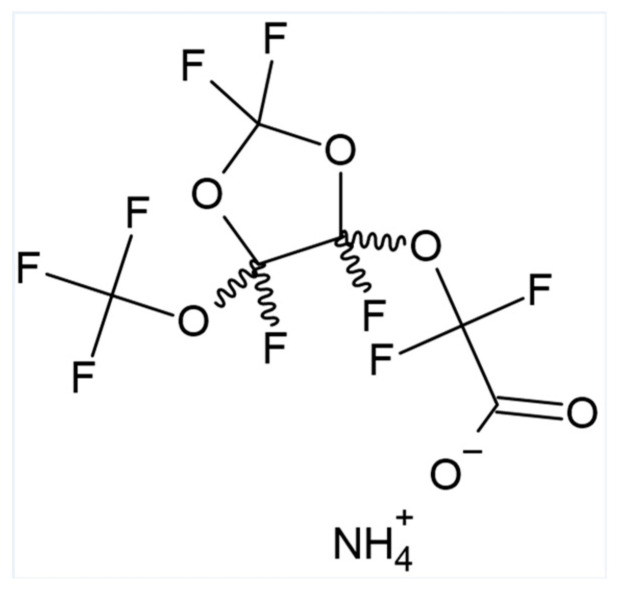
Molecular structure of cC_6_O_4_.

**Figure 2 toxics-11-00284-f002:**
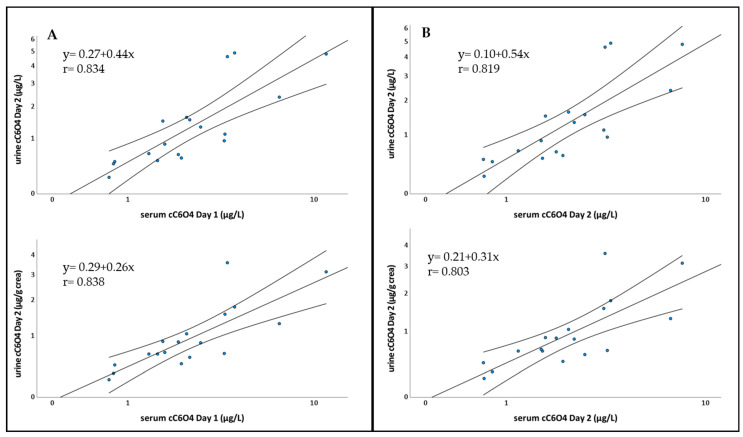
Correlation between the concentration of cC_6_O_4_ in day 2 urine not adjusted for creatinine(above) and adjusted for creatinine (below) in day 1 serum (**A**), or day 2 serum (**B**).

**Figure 3 toxics-11-00284-f003:**
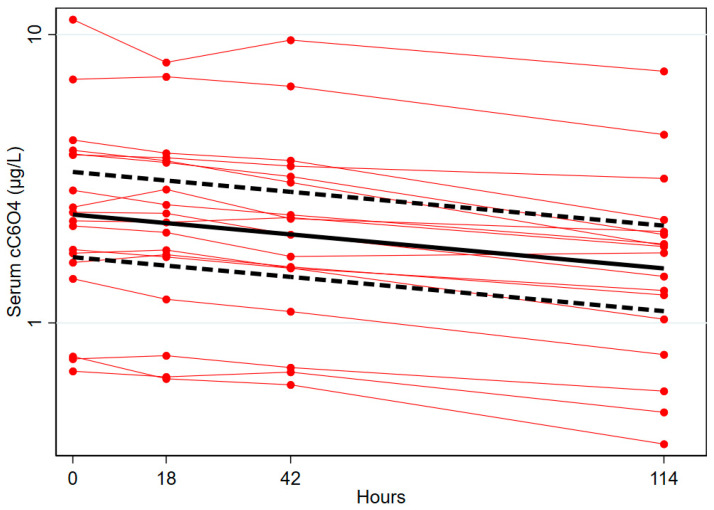
Elimination of cC_6_O_4_ from serum. Data from each individual are reported separately with red dots connected by red lines, while the linear regression and 95% confidence interval are reported in bold black continuous and dotted lines.

**Table 1 toxics-11-00284-t001:** Summary of characteristics of study subjects, available samples and urinary creatinine.

	Parameter (Unit)	Statistics	Data
Personal characteristics	Male subjects	n (%)	18 (100)
	Age, year	mean ± SD	44 ± 5
	Weight, kg	mean ± SD	79 ± 6
	Height, cm	mean ± SD	175 ± 6
	BMI, kg/m^2^	mean ± SD	26.1 ± 2.3
	Current tobacco smoker	n (%)	4 (22)
	e-cig smoker	n (%)	1 (5)
	Use of drugs	n (%)	14 (78)
Available samples	Blood samples	n	72
	Blood sample/subject	n	4
	Urine samples	n	254
	Urine sample/subject (day 1–day 3)	n, mean ± SD	14 ± 4
	Lost urine samples	n	14
Urinary creatinine	Creat (g/day)	mean ± SD	1.465 ± 0.401
	Expected Crea (g/day)	mean ± SD	1.905 ± 0.144
	Crea ratio (%)	mean ± SD	77 ± 21
	Subjects with crea ratio < 60%	n (%)	4 (22)

**Table 2 toxics-11-00284-t002:** Concentration of cC_6_O_4_ in serum and urine for each sampling day.

ID Subject	cC_6_O_4_ Serum µg/L				cC_6_O_4_ Urine µg/L				cC_6_O_4_ Urine µg/g Crea			
	Day 1	Day 2	Day 3	Day 6	Day 1	Day 2	Day 3	Day 6	Day 1	Day 2	Day 3	Day 6
1	0.68	0.65	0.68	0.49	0.20	0.23	0.19		0.30	0.22	0.22	
2	0.77	0.64	0.61	0.38	0.71	0.50	0.49		0.48	0.44	0.43	
3	1.62	1.72	1.57	1.25	0.51	0.52	0.42		0.65	0.63	0.50	
4	0.75	0.77	0.70	0.58	0.61	0.46	0.38		0.32	0.31	0.26	
5	2.17	2.06	1.70	1.75	0.63	0.64	0.66		1.10	0.87	0.71	
6	2.89	2.57	2.37	1.88	1.76	1.32	1.47	1.95	1.11	0.85	0.37	0.46
7	2.52	2.91	2.30	2.08	1.60	1.54	1.76	0.68	0.56	0.57	0.55	0.23
8	2.42	2.40	2.03	1.45	1.45	1.62	1.71	1.43	1.25	1.05	1.02	0.78
9	2.26	2.24	2.32	1.84	0.45	0.57	0.70	0.59	0.60	0.46	0.23	0.33
10	1.75	1.79	1.55	1.03	1.06	1.50	0.64	0.39	0.87	0.88	1.55	0.44
11	11.29	8.01	9.57	7.46	4.20	4.82	4.02	3.48	3.73	3.14	1.89	1.63
12	1.80	1.69	1.55	1.30	0.64	0.87	0.60	0.50	0.63	0.66	0.43	0.35
13	3.97	3.65	3.07	1.86	3.89	4.63	6.08	3.76	5.12	3.59	2.59	1.18
14	1.42	1.21	1.09	0.78	0.70	0.66	0.68	0.58	0.59	0.63	0.32	0.44
15	3.83	3.74	3.50	3.17	0.78	0.95	1.09	0.65	0.49	0.64	0.93	0.47
16	3.86	3.60	3.22	2.03	2.40	1.12	1.45	2.09	1.60	1.56	1.49	0.97
18	4.31	3.88	3.66	2.28	5.92	4.90	2.48	3.12	2.11	1.78	1.49	1.27
19	7.00	7.15	6.62	4.50	2.45	2.38	1.71	2.94	2.19	1.30	1.48	1.65
Mean	3.07	2.82	2.67	2.01	1.66	1.62	1.47	1.70	1.32	1.09	0.91	0.78
SD	2.58	2.03	2.24	1.69	1.57	1.55	1.48	1.26	1.28	0.93	0.69	0.50
Min	0.68	0.64	0.61	0.38	0.20	0.23	0.19	0.39	0.30	0.22	0.22	0.23
p25	1.62	1.69	1.55	1.03	0.63	0.57	0.60	0.59	0.56	0.57	0.37	0.44
Median	2.34	2.32	2.17	1.80	0.92	1.04	0.90	1.43	0.76	0.76	0.63	0.47
p75	3.86	3.65	3.22	2.08	2.40	1.62	1.71	2.94	1.60	1.30	1.49	1.18
Max	11.29	8.01	9.57	7.46	5.92	4.90	6.08	3.76	5.12	3.59	2.59	1.65

**Table 3 toxics-11-00284-t003:** Amount of cC_6_O_4_ in plasma and urine and % of urinary excretion calculated considering the amount of cC_6_O_4_ in urine of day 2 vs. the amount in serum of day 1 or day 2.

	cC_6_O_4_ in Serum Day 1 µg	cC_6_O_4_ in Serum Day 2 µg	cC_6_O_4_ in Urine Day 2 µg	Urinary Excretion (vs. Serum Day 1) %	Urinary Excretion (vs. Serum Day 2) %
**Mean**	9.50	8.73	1.62	18.9	20.1
**SD**	7.95	6.36	1.42	10.1	11.0
**Min**	1.92	1.70	0.34	4.8	4.9
**p25**	5.51	5.38	0.74	12.5	12.8
**Median**	7.25	7.76	1.05	17.5	17.9
**p75**	10.92	10.42	1.85	24.1	25.7
**Max**	34.08	24.18	5.37	46.4	50.5

**Table 4 toxics-11-00284-t004:** cC_6_O_4_ renal clearance (Cl) and distribution volume (Vd).

Statistics	Cl mL/min	Cl_w_ mL/min × kg	Serum Volume L	Vd, L	Vd_w_ L/kg
Mean	0.41	0.0052	3.1	6.6	0.084
SD	0.21	0.0026	0.4	3.4	0.042
Min	0.13	0.0015	2.6	2.1	0.023
p25	0.29	0.0037	2.9	4.6	0.06
Median	0.38	0.0051	3.0	6.1	0.081
p75	0.53	0.0064	3.1	8.6	0.103
Max	1.02	0.0128	4.0	16.3	0.204

**Table 5 toxics-11-00284-t005:** Summary of studies investigating the half-life of some PFAS in occupationally exposed workers.

Chemical	Study	Study Subjects/Company/Country	N Subjects	N Samples, N/Subject	Years of Study	C_0_ Serum µg/L	t_1/2_
PFOA	[5]	Fluorochemical workers after the ban of PFOA; Solvay, Italy	93 (15 F)	568, 2 to 9/subject	2013–2021	Median 750	GM 3.16 (95% CI 2.98–3.37) years
PFOA	[17]	Retired fluorochemical workers; 3M, US	26 (2 F)	4 to 8/subject	1998–2004	Mean 691	GM 3.5 (95% CI 3.0–4.1) years
PFOA	[4]	Retired fluorochemical workers; Miteni, Italy	35 M	81 2/subject	2000–2018	GM 1.489	Mean 3.35 (95% CI 2.89–3.99) years
Perfluorobutyrate (PFBA)	[14]	Fluorochemical workers removed from the workplace for >7 days; 3M, US	9 (2 F)	2/subject	2007	2–71	72 (95% CI 43–101) h
GenX	[15,16]	Fluorochemical workers off-work weekend (no exposure for 3–4 days); Chemours, NL	18	2/subject	Before 2021	Not reported	Mean ± SD 81 ± 55 h
cC_6_O_4_	Present study	Fluorochemical workers removed from the workplace for 5 days; Solvay, Italy	18 M	72 4/subject	2021–2022	Median 2.34	GM 184 (95% CI 162–213) h

## Data Availability

The data presented in this study are available on request from the corresponding author. The data are not publicly available due to privacy reasons.

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
