# Peer review of "Kinetics of Excretion of the Perfluoroalkyl Surfactant cC6O4 in Humans"

_toxics, 2023, doi:10.3390/toxics11030284_

Round 1

Reviewer 1 Report

General Comments:

The study was designed to study the elimination kinetics in occupational populations exposed to cC6O4. Through the investigation of cC6O4 in exposed individuals in occupational populations, the half-life of cC6O4 in the blood of the population sample and the correlation between blood and urine content were calculated, and the result was that cC6O4 had a shorter half-life of 7.7 days than traditional PFAS. However, the manuscript needed some modifications before it could be published.

Comments:

1. How to exclude the influence of diet or other factors in life on the cC6O4 in the human body?

2. How do authors control for individual susceptibility factors in occupational populations?

3. The sample size of the survey population is a bit small

4. Can animal tests be added to further verify the elimination kinetics of cC6O4?

5. How do the authors consider the elimination kinetics of cC6O4 in other, younger age groups?

Author Response

REVIEWER 1 – TOXICS

Comments and Suggestions for Authors

General Comments:

The study was designed to study the elimination kinetics in occupational populations exposed to cC6O4. Through the investigation of cC6O4 in exposed individuals in occupational populations, the half-life of cC6O4 in the blood of the population sample and the correlation between blood and urine content were calculated, and the result was that cC6O4 had a shorter half-life of 7.7 days than traditional PFAS. However, the manuscript needed some modifications before it could be published.

Comments:

  1. How to exclude the influence of diet or other factors in life on the cC6O4 in the human body?

ANSWER TO COMMENT 1. We tested a group of individuals of the general population, during the development of the analytical assay. We could appreciate that cC6O4 was not present in these subjects, as you can see in the publication:

  • Frigerio G, Cafagna S, Polledri E, Mercadante R, Fustinoni S. Development and validation of an LC-MS/MS method for the quantitation of 30 legacy and emerging per- and polyfluoroalkyl substances (PFASs) in human plasma, including HFPO-DA, DONA, and cC6O4. Anal Bioanal Chem. 2022 Jan;414(3):1259-1278. doi: 10.1007/s00216-021-03762-1. Epub 2021 Dec 15. Erratum in: Anal Bioanal Chem. 2022 Mar;414(6):2315.

Available at: https://link.springer.com/article/10.1007/s00216-021-03762-1

Following this result, we assume the diet and other lifestyle factors do not affect the presence of this chemical in the human body.

  1. How do authors control for individual susceptibility factors in occupational populations?

ANSWER TO COMMENT 2.We did not control for individual susceptibility factors. We recognize that they may affect the variability of the kinetics, but this was taken into account by the interval of confidence of the half-life of cC6O4.

  1. The sample size of the survey population is a bit small

ANSWER TO COMMENT 3. We agree with the reviewer, but it is a typical sample size of a kinetics study in occupationally exposed individuals. See similar experiences in the following papers, cited in our manuscript:

  • Chang, S.C.; Das, K.; Ehresman D.J.; Ellefson, M.E.; Gorman, G.S.; Hart, J.A.; Noker, P.E.; Tan, Y.M.; Lieder, P.H.; Lau, C; Olsen, G.W.; Butenhoff, J.L. Comparative pharmacokinetics of perfluorobutyrate in rats. mice. monkeys. and humans and relevance to human exposure via drinking water. Toxicol Sci. 2008 Jul;104(1):40-53. doi: 10.1093/toxsci/kfn057.
  • Arbo Unie. 2020. Study Half-Time HFPO-DA. Arbo Unie, Utrecht, the Netherlands. https://heronet.epa.gov/heronet/index.cfm/reference/details/reference_id/8631852., accessed 18th May, 2022
  • Clark, D.S. The Chemours Company. 2021, March 17. Letter to EPA, Office of Pollution Prevention and Toxics regarding propanoic acid, 2,3,3,3-tetrafluoro-2-(1,1,2,2,3,3,3-heptafluoropropoxy)-CAS RN 13252-13-6 (also known as HFPO-DA). https://heronet.epa.gov/heronet/index.cfm/reference/details/reference_id/8631852, accessed 6th Jun, 2022
  • S. Environmental Protection Agency, Human Health Toxicity Values for Hexafluoropropylene Oxide (HFPO) Dimer Acid and Its Ammonium Salt (CASRN 13252-13-6 and CASRN 62037-80-3), EPA Document Number: 822R-21-010, accessed 2nd Oct 2021.
  1. Can animal tests be added to further verify the elimination kinetics of cC6O4?

ANSWER TO COMMENT 4.The animal tests were performed for the preparation of the dossier for registration to ECHA. The data are publicly available on the ECHA site at:

ECHA, Registration dossier, Acetic acid, 2,2-difluoro-2-[[2,2,4,5-tetrafluoro-5-(trifluoromethoxy)-1,3-dioxolan-4-yl]oxy]-, ammonium salt (1:1)). According to CLP (Regulation (EC) No 1272/2008; link: https://echa.europa.eu/it/registration-dossier/-/registered-dossier/5712, accessed Oct 10th, 2022

The data is cited in the introduction and the reference to the document is given:

Indeed, toxicokinetics studies in rats provided evidence of a short half-life in se-rum (4 to 7 hrs), with the majority of the excretion via urine occurring usually within 24 hours, and no evidence of biotransformation and bioaccumulation [1]

  1. How do the authors consider the elimination kinetics of cC6O4 in other, younger age groups?

ANSWER TO COMMENT 5. Our study was performed investigating adult males occupationally exposed. We recognize that this cannot always be the case; following this comment, we added a caution sentence to highlight that other groups could have different kinetics parameters.

“As this study does not includes females and younger or elder subjects, differences in the kinetics parameters could be present in individuals with different personal characteristics”.

Reviewer 2 Report

This is a careful study with a well-planned strategy and sound analytical technique.   Unfortunately, at times the English is a bit tortuous making it difficult to follow eg. “Comparing results of serum cC6Oreported here with the previous work [5], we note a lowering of both median and maximum concentrations from 4 and 65 μg/L in Plastomer 2, and 4 and 20 μg/L in Plastomer 1 departments in 2021, to 2.34 and 11.3 μg/L in this study.”

L260.  It is unclear how the kinetics of elimination was calculated.   Was linear regression applied to each individual separately and then the rates averaged over all subjects or was linear regression applied to all the pooled data?   If the latter, how is the variable intercept taken into account?

L281 This statement should be a conclusion not the start of the discussion.  “In the present work for cC6Oa half-life of 7.7 (95% CI 6.8-8.9) days was estimated. The amount of cC6Oexcreted unchanged in urine suggests that the molecule is not metabolised and that urine is the solely excretion route.”

L321.  Difficult to follow – please simplify. “In the study of the kinetics of excretion of cC6Ousing the linear regression betweenln-transformed concentrations of cC6Ovs. time, the first issue is influencing the intercept, but not the slope of the linear regression, finally not affecting the main result of the kinetics study.”

L347.  Don’t understand how the half-lives of ~7 days is comparable to ~3 years.  “Noticeably, the half-lives 347 of these compounds were all found to be within a few days (3 to 7.7 days), comparedto the 3.16 - 3.5 years reported for PFOA [4, 5, 18].”

Author Response

REVIEWER 2 – TOXICS

Comments and Suggestions for Authors

This is a careful study with a well-planned strategy and sound analytical technique.   Unfortunately, at times the English is a bit tortuous making it difficult to follow eg. “Comparing results of serum cC6O4 reported here with the previous work [5], we note a lowering of both median and maximum concentrations from 4 and 65 μg/L in Plastomer 2, and 4 and 20 μg/L in Plastomer 1 departments in 2021, to 2.34 and 11.3 μg/L in this study.”

ANSWER TO COMMENT 1. Thank you for your favourable comments. We rephrased the sentence to improve its clarity.

L260.  It is unclear how the kinetics of elimination was calculated.   Was linear regression applied to each individual separately and then the rates averaged over all subjects or was linear regression applied to all the pooled data?   If the latter, how is the variable intercept taken into account?

ANSWER TO COMMENT 2.The linear regression was applied to all the pooled data. This was better specified in the text.

L281 This statement should be a conclusion not the start of the discussion.  “In the present work for cC6O4 a half-life of 7.7 (95% CI 6.8-8.9) days was estimated. The amount of cC6O4 excreted unchanged in urine suggests that the molecule is not metabolised and that urine is the solely excretion route.”

ANSWER TO COMMENT 3.Following this comment, we rephrased the first sentence of the discussion.

L321.  Difficult to follow – please simplify. “In the study of the kinetics of excretion of cC6O4 using the linear regression betweenln-transformed concentrations of cC6O4 vs. time, the first issue is influencing the intercept, but not the slope of the linear regression, finally not affecting the main result of the kinetics study.”

ANSWER TO COMMENT 4.Thank you for your comment. We rephrased the sentence and simplified it.

L347.  Don’t understand how the half-lives of ~7 days is comparable to ~3 years.  “Noticeably, the half-lives 347 of these compounds were all found to be within a few days (3 to 7.7 days), comparedto the 3.16 - 3.5 years reported for PFOA [4, 5, 18].”

ANSWER TO COMMENT 5.We agree with the reviewer that the half-lives of the new perfluorinated alkyl surfactants are not comparable with the half-live of PFOA; the sentence has been rephrased.

Reviewer 3 Report

The manuscript titled, "Kinetics of excretion of the perfluoroalkyl surfactant cC6O4 in humans" describe experiments related to the evaluation of excretion kinetics of the surfactant in occupationally exposed workers. In general, the manuscript is easy to read; however further improvements written language style is required before publication. 

The reviewer agrees with the methodology used by the authors; however, he is not very convinced by the approach the authors took to estimate the completeness of urine collection (using "expected" creatinine amount). Some re-thinking is required. 

My detailed comments are attached herewith.

Author Response

REVIEWER 3- TOXICIS

Comments and Suggestions for Authors

The manuscript titled, "Kinetics of excretion of the perfluoroalkyl surfactant cC6O4 in humans" describe experiments related to the evaluation of excretion kinetics of the surfactant in occupationally exposed workers. In general, the manuscript is easy to read; however further improvements written language style is required before publication. 

The reviewer agrees with the methodology used by the authors; however, he is not very convinced by the approach the authors took to estimate the completeness of urine collection (using "expected" creatinine amount). Some re-thinking is required. 

ANSWER TO COMMENT. We agree with the reviewer that the approach to estimate completeness of urine collection using expected creatinine amount is affected by a large degree of uncertainty. Following this comment we changed the verb “calculate” with “estimate”. Furthermore, we recognise that this estimate can be affected by a large degree of variability (± 40 %) (see L 198-200). For taking under control missing voids, we were also using a urine collection diary, as suggested by the reviewer in the notes. However, we could verify that not all the missed voids were recorded by study subjects. Therefore, we conclude that neither the approaches could really give a complete and real picture of the completeness of urine collection. Nevertheless, we used either estimates to make some reasonable considerations.

My detailed comments are attached herewith.

We answered all detailed comments directly into the manuscript. We have uploaded it here.
